# The Moderating Roles of Intrinsic and Extrinsic Religiosity on the Relationship between Social Networks and Flourishing: A Study on Community-Dwelling Widowed Older Adults in Malaysia

**DOI:** 10.3390/healthcare11091300

**Published:** 2023-05-02

**Authors:** Hui Foh Foong, Tengku Aizan Hamid, Rahimah Ibrahim, Mohamad Fazdillah Bagat

**Affiliations:** 1Malaysian Research Institute on Ageing (MyAgeingTM), Universiti Putra Malaysia, Serdang 43400, Selangor, Malaysia; 2Department of Human Development and Family Studies, Faculty of Human Ecology, Universiti Putra Malaysia, Serdang 43400, Selangor, Malaysia

**Keywords:** flourishing, social networks, intrinsic religiosity, extrinsic religiosity, widowhood

## Abstract

Widowhood affects the social networks and well-being of older adults. Religion might moderate the relationship between a stressor and well-being. This study aimed to identify the moderating roles of intrinsic and extrinsic religiosity on the relationship between social networks and flourishing among widowed older people and whether this relationship varied across gender. This study involved 655 community-dwelling widowed older Malaysians from Wave 1 (2012–2013) of “Identifying Psychosocial Risks and Quantifying the Economic Costs of Age-Related Cognitive Decline among Older Malaysians” in Peninsula Malaysia. The moderated hierarchical multiple linear regression analysis was conducted to examine the moderating roles of religiosity. Results showed that the moderating effect of religiosity on the relationship between social networks and flourishing was only observed for extrinsic religiosity, not intrinsic religiosity. In terms of gender differences, extrinsic religiosity moderated the relationship between social networks, flourishing only among widows but not widowers. Widows with low levels of extrinsic religiosity should join activities or programs that could expand their social networks to promote higher well-being despite widowhood.

## 1. Introduction

Psychological well-being and health are closely associated. Therefore, the link between these two variables is essential in older adults as the prevalence of morbidity, multimorbidity, and disability increases with advancing age. Besides that, due to the advancement of medical technology, life expectancy has increased, ultimately making the issue of psychological well-being in later life a growing concern. According to Lukaschek and colleagues, being a woman, poverty, a sedentary lifestyle, multimorbidity, depression, anxiety, and sleeping disorders were associated with low psychological well-being in later life [1]. Additionally, Steptoe and colleagues also noticed a two-way relationship between physical health and subjective well-being in older adults [2]. Furthermore, according to a systematic review, there was a consensus that higher religiosity was associated with lower depressive symptoms, suicidal ideation, anxiety, and stress [3]. The positive relationship between religion and mental health could be attributed to stronger coping skills and social networks [3]. 

Older adults are more prone to widowhood, which involves many economic, social, and psychological difficulties [4]. Since women live longer than men, older women tend to experience spousal loss. Spousal loss is a major life event that could result in physical and mental health problems, the latter causing more long-term effects [5,6,7]. According to a systematic review, the prevalence of depression was highest within the first month of widowhood, at about 40%. It decreased within the first five years of widowhood to a prevalence of about 10% [8]. Additionally, the emotional isolation that follows the loss of a spouse cannot be overcome with kin-based social support and social integration, contributing to worsening mental health [9]. Further, social relationships within the first year of widowhood do not aid much in the recovery process of widows or widowers [10]. Widowhood is mostly viewed as a female-specific issue, as women are more prone to widowhood compared to men. The reason is that women possess a higher life expectancy and a higher tendency to remain unmarried upon the demise of their spouse [11]. 

Positive psychology describes a relatively novel concept known as flourishing. Flourishing differs from other measurements of positive psychology as it considers both hedonic and eudemonic aspects of well-being [12]. Flourishing among human beings represents a mental state in which an individual actively strives to improve living standards instead of merely feeling good [13]. Investigation into flourishing in old age is crucial considering how flourishing has been linked to a range of psychological and physical health benefits, especially cardiovascular health [14,15]. Moreover, absence or low levels of flourishing were associated with an increased probability of all-cause mortality for both men and women of all ages, even after adjustment for known causes of death [16]. While a recent study reported a negative correlation between widowhood and the subjective well-being of older adults, the relationship was partially attributed to the reduced frequency of leisure activity [17]. A previous study in Malaysia showed that demographic variables such as being male, having employment, and having living children were associated with higher levels of flourishing in older adults [12]. Furthermore, the demographic predictors of flourishing varied between rural and urban areas, with being female, having higher levels of education, being employed, and having larger family sizes associated with higher flourishing in rural areas. However, the predictors above were not significant among older people living in urban areas [18]. 

Family and friends’ social networks are significant as a source of support, especially in later life. Additionally, it has been associated with higher life satisfaction [19], better health outcomes [20], and a lower risk of premature mortality [21]. However, studies revealed that the death of a spouse could affect social networks, social ties, and social support [5,22], possibly due to the loss of an intimate partner with whom one used to share everything, as well as losing touch with those who used to be close to the deceased. The death of a spouse causes an unwarranted reorganization of social networks to be incorporated and adapted [23]. According to Guiaux and colleagues, older adult connections and support decrease significantly approximately 2.5 years post-widowhood [24]. Becker and colleagues reported that marriage was positively correlated with well-being and negatively correlated with depressive symptoms in both older men and women [25]. They also observed that all types of network characteristics, such as size, closeness, contact frequency, and proximity, were correlated with well-being and depressive symptoms in older people [25]. The positive effects of social networks on well-being among widowed older adults could be attributed to the moderating influences of the environment (stressors) on well-being [26].

Religious orientation can be divided into intrinsic religiosity and extrinsic religiosity [27]. Intrinsic religiosity refers to personal and internalized spiritual beliefs, attitudes, or values towards religion, spirituality, or its practices. In contrast, extrinsic religiosity refers to religion as “an instrumental means to solace and sociability”, often involving religious memberships, social activities, and/or religious preferences [27]. Extensive literature has shown the positive role of religiosity in promoting mental health. Positive life appraisal was higher among those with high religious and spiritual involvement [7,28,29]. Case in point, religiosity was found to moderate the relationship between social isolation and psychological well-being among older Malay Muslims [30]. A recent study by Foong and colleagues found that intrinsic religiosity was positively associated with life satisfaction among older adults living with chronic diseases [31]. The potential moderating effects of intrinsic religiosity on the relationship between depression and cognitive function among community-dwelling older adults were investigated [32]. Religiosity and spirituality were also found to be important among widows [7]. A review found that religious coping, as well as religious/spiritual beliefs and behaviors, were used to facilitate positive adjustment to the loss of a spouse among older women [33]. 

Pargament’s Theory of Religious Coping guided this study. In the combined religious moderator–deterrent model, Pargament suggested that religious coping could mediate a stressor and stress level, as religious coping could protect religious people from the negative impacts of stress. Moreover, religious coping is a strong predictor of more favorable outcomes, regardless of stress intensity [34]. Previous reports have established the moderating effects of religiosity on the relationship between social isolation and psychological well-being in older people [30]. However, there appears to be a scarcity of gerontological studies focused on the moderating effects of religiosity in the relationship between social networks and flourishing, as well as associated gender differences, particularly among older widows and widowers. This study is intended to observe gender differences as part of the moderation model being tested due to reports from past studies denoting gender differences in flourishing and social networks. To elaborate further, older men have shown higher levels of flourishing than older women [12]. In addition, older women were found to have reduced social resources, such as smaller social networks, compared to their male counterparts [35,36], as traditional gender roles dictate that women are expected to be housebound. Studies further corroborate that the negative impact of widowhood is more prominent in women due to financial strains upon the loss of a spouse and healthcare system inequalities [37]. As such, an examination of gender differences could inform policymakers about the need to develop gender-specific interventions to improve psychological well-being among widowed older adults. 

Religiosity might be a potential source of well-being among widowed older people with small social networks and that information on the relationships between social networks, flourishing, and religiosity in widowed older people specifically is limited. To this end, this study aimed to: (1) examine the relationship between social networks and flourishing in widowed older adults; (2) examine the moderating roles of intrinsic and extrinsic religiosity on the relationship between social networks and flourishing; and (3) investigate the gender differences in the moderation models. A review of the aforementioned literature led to the postulation of three hypotheses: H_1_A—There is a significant positive relationship between social networks and flourishing; H_2_A—As the value of intrinsic religiosity increases, the relationship between social networks and flourishing decreases; H_3_A—As the value of extrinsic religiosity increases, the association between social networks and flourishing decreases; and H_4_A—The relationships between social networks, intrinsic religiosity, extrinsic religiosity, and flourishing vary according to gender. 

## 2. Materials and Methods

### 2.1. Data and Sampling

This study was a secondary data analysis, and data were drawn from Wave 1 (2012–2013) of “Identifying Psychosocial Risks and Quantifying the Economic Costs of Age-Related Cognitive Decline among Older Malaysians” in Peninsula Malaysia. This study was a two-wave longitudinal study; however, only baseline data were used in the present study. The original survey consisted of 2322 adults aged over 60. The sample size calculation was determined using the formula for a study estimating population prevalence. The sample size was determined based on the expected prevalence of diseases or health-related problems in the population, a margin of error, and confidence intervals. The inclusion criteria for this study were adults aged 60 or older, Malaysian nationality, absence of severe mental disorders such as dementia, and consent to participate in the study. Older adults with severe mental disorders were excluded from this study as they could not provide consent and might not respond well to the questionnaire provided. The sample size required was 2800, and 2322 respondents participated in the study, with a response rate of 82.9%. Details of this study, including design, sampling, and sample size calculation, have been described elsewhere [38]. The same dataset has also been used in other publications [39,40]. A total of 655 (28.2% of the original sample) respondents in the subsample for this study were widows/widowers. The survey covered a wide range of topics. However, only data relevant to the study’s variables of interest (i.e., sociodemographic and economic, medical history, flourishing, religiosity, and social networks) were utilized. The survey employed a multi-stage stratified random sampling technique. Marital status was assessed based on the participant’s response to a query on whether they were single (never married), married, separated/divorced, or widowed/widower. 

### 2.2. Flourishing (Dependent Variable)

Flourishing was measured via the Flourishing Scale developed by Diener and colleagues [41]. The scale consisted of eight items on a 7-point Likert-type scale ranging from 1 (strongly agree) to 7 (strongly disagree). Examples of items include “I lead a purposeful and meaningful life”, “I actively contribute to the happiness and well-being of others”, and “I am optimistic about my future”. Scores ranged from 8 to 56, where higher scores indicate higher levels of well-being. The validity of the scale among older people has been established elsewhere and found to be suitable for administering to Malaysian older adults [12]. Overall, the scale demonstrated excellent internal consistency with Cronbach’s α = 0.94. Flourishing scores were treated as a continuous variable during descriptive analysis, Pearson’s correlation, and moderated multiple regression analysis. 

### 2.3. Social Networks (Independent Variable)

The Lubben Social Network Scale-6 was used to measure social networks [42]. The instrument is a validated instrument used to evaluate social isolation in older adults by measuring the number and frequency of social contact with friends and family members, as well as the perceived social support received from these sources. Possible scores ranged from 0 to 30, with higher scores indicating a larger social network. The validity of this scale among older people has been established [43]. Scores of 12 and below indicated a risk of social isolation [42] and were used as a cut-off point to identify the prevalence of older adults at risk of social isolation during descriptive analysis. The scale was found to be feasible for administering to the local older population [44]. Overall, the scale demonstrated good internal consistency with Cronbach’s α = 0.77. Social network scores were treated as a continuous variable during Pearson’s correlation and moderated multiple regression analysis.

### 2.4. Intrinsic and Extrinsic Religiosity (Moderating Variables)

Intrinsic and extrinsic religiosity were measured via the revised intrinsic/extrinsic religious orientation scale [45]. The scale consists of 14 items on a five-point Likert scale ranging from 1 (strongly disagree) to 5 (strongly agree). Six items measured intrinsic religiosity, while the remaining eight assessed extrinsic religiosity [45]. Higher scores indicated greater levels of intrinsic and/or extrinsic religiosity. The validity of this scale for adults has been proven elsewhere [46]. The scale has been used in past studies among Malaysian older adults and found to be feasible [31,32]. Overall, the scale demonstrated good internal consistency with Cronbach’s α = 0.76. Intrinsic and extrinsic religiosity scores were treated as continuous data during descriptive analysis, Pearson’s correlation, and moderated multiple regression analysis. 

### 2.5. Sociodemographic, Economic, and Health-Related Characteristics (Covariates)

Background variables such as age, gender, year(s) of education, employment status, poverty status, ethnicity, multimorbidity status, and living arrangements were controlled for in the 4-step moderated hierarchical multiple linear regression. Poverty status was identified through self-reported household income, with household income less than MYR 460 categorized as hardcore poor [47]. Multimorbidity status was examined via self-reported medical history. The assessment included 12 chronic diseases [31]. The presence of more than one chronic disease was classified as multimorbidity. Categorical variables were dummy-coded for regression analysis: gender (male = 0, female = 1), employment status (currently employed = 0, currently not employed = 1), poverty status (nonhardcore poor = 0, hardcore poor = 1), living arrangement (living with others = 0, living alone = 1), multimorbidity status (multimorbidity = 1, no multimorbidity = 0), and ethnicity (Chinese and Indian = 0, Malay = 1). Age and year(s) of education were treated as continuous variables. 

### 2.6. Analytic Strategy, Data Preparation, and Confirmatory Factor Analysis (CFA)

The original dataset consisted of 2322 respondents. However, current analyses involve widows and widowers, resulting in a total of 655 respondents (accounting for 28.2% of the original dataset). No missing values were found for gender, living arrangement, multimorbidity status, age, or year(s) of education. However, there were missing values in employment status (2.1%), ethnicity (0.3%), hardcore poor status (4.0%), flourishing (3.5%), intrinsic religiosity (2.9%), extrinsic religiosity (4.0%), and social networks (2.1%) from the 655 responses. The pairwise deletion was used in the handling of missing data, given the overall missing data percentage of less than 10% and missing completely at random pattern [48].

The data analysis for this study was conducted in three stages. Firstly, a univariate analysis was carried out to describe the sample according to gender. Secondly, bivariate correlations were analyzed to assess the relationships between the independent variables and the dependent variable. Finally, a 4-step moderated hierarchical multiple linear regression model was developed. This involved regressing flourishing on social networks, the key independent variable, and moderating variables (intrinsic and extrinsic religiosity), while controlling for relevant covariates. The moderated hierarchical multiple linear regression analysis involved four steps, beginning with the inclusion of covariates, followed by the inclusion of independent variables and moderators, respectively. The fourth and final step involved the inclusion of interaction terms. A significant interaction term indicated the presence of a moderating effect. This study complied with Robinson and colleagues’ suggestion to conduct a post-hoc investigation to observe the significance of differences in simple slopes given the presence of moderation effects [49]. Next, full sample and gendered-stratified analyses were conducted, respectively, to provide an overall assessment and to observe potential gender differences in the moderating effects of intrinsic and extrinsic religiosity on the relationship between social networks and flourishing. A *p*-value of less than 0.05 was deemed statistically significant, and all analyses were performed using SPSS v.26.0 (IBM, Armonk, NY, USA).

A general concern that follows moderating analysis is multicollinearity, as it could distort the beta coefficient. However, this could be prevented by centering. The continuous independent variable and moderator were therefore centered before creating interaction terms. Tolerance values ranged from 0.667–0.970 for Model 1, 0.803–0.986 for Model 2, and 0.719–0.910 for Model 3, all of which were greater than 0.2, indicating no presence of multicollinearity [50].

Pooled confirmatory factor analysis (CFA) was performed using the AMOS Graphic version 24 to establish the construct validity of the measurement model of social networks, religiosity, and flourishing among the targeted population before subsequent data analysis and interpretation. Evidence of construct validity was established by evaluating the Tucker–Lewis index (TLI), comparative fit index (CFI), and root-mean-square error of approximation (RMSEA). The model goodness-of-fit indices of the pooled measurement model for wave one were TLI = 0.931, CFI = 0.945, and RMSEA = 0.074. Next, the model goodness-of-fit indices of the pooled measurement model for wave two were TLI = 0.939, CFI = 0.952, and RMSEA = 0.060. TLI and CFI values were all > 0.90, and RMSEA values were <0.08 in the first and second waves, indicating an acceptable absolute and incremental fit [51]. 

In addition, an absence of measurement bias against male and female groups is a prerequisite for the subsequent analysis. Therefore, pairwise, multigroup pooled CFA with robust maximum likelihood estimation from wave two data was used to examine configural and scalar invariances [52]. All factor loadings and thresholds were freely estimated without constraints in the configural invariance test [52,53]. Results showed that the configural invariance model fit the data well (CFI = 0.971, TLI = 0.963, RMSEA = 0.036), indicating the basic organization of measurements was supported in male and female groups. Next, we checked the scalar invariance by constraining the item intercepts to be equivalent in the male and female groups. Again, the scalar invariance model fit the data well (CFI = 0.947, TLI = 0.937, RMSEA = 0.047), indicating a strong invariant of measurements across male and female groups. In sum, these results suggest that the social networks, religiosity, and flourishing measurements used in this study did not differ across different gender. 

## 3. Results

### 3.1. Descriptive Analysis of Independent and Dependent Variables Based on Gender

This study involved a total of 655 community-dwelling older adults who had been widowed. Table 1 depicts a descriptive analysis of independent and dependent variables according to gender. The sample consisted mostly of women (85.2%) compared to men (14.8%), which is expected given their higher life expectancy and the tendency for women to marry older men. The participants had a mean age of 71.4 ± 6.79 years andmajority were Malay (68.6%). Men were significantly older than women at a *p* < 0.001 level, and most of them lived with others (73.1%). In terms of socio-economic status, respondents attended formal education for an average of 3.5 ± 3.53 years, most of them within the currently not working (85.5%) and non-hardcore poor (53.7%) categories. Men had significantly higher levels of education than women at the *p* < 0.001 level, and more men fell under the current working category as compared to women (*p* = 0.023). However, no association was found between poverty status and gender. Men reported significantly larger social networks than women at the *p* < 0.05 level; however, no significant gender differences were observed in terms of flourishing, intrinsic religiosity, or extrinsic religiosity. Participants with multimorbidity (49.9%) were almost equal to those without multimorbidity (50.1%), and no association was found between gender and multimorbidity status. 

### 3.2. Correlation among Study Variables 

Pearson correlations (Table 2) were administered to provide insight into the relationships between variables of interest. Flourishing was significantly correlated to age (r = −0.097, *p* = 0.015), year(s) of education (r = 0.091, *p* = 0.022), ethnicity (r = 0.201, *p* < 0.001), social networks (r = 0.238, *p* < 0.001), intrinsic (r = 0.418, *p* < 0.001), and extrinsic religiosity (r = 0.222, *p* < 0.001). 

### 3.3. Moderating Effects of Intrinsic and Extrinsic Religiosity on the Relationship between Social Networks and Flourishing in Overall Sample, Men, and Women

A 4-step moderated hierarchical multiple regression was employed to identify if intrinsic religiosity moderated the association between social networks and flourishing in the overall sample, as well as to observe gender variation. Table 3 depicts the beta coefficient, standard error, and *p*-value in the overall, widower, and widow models. All models, upon controlling for possible covariates, found social networks and intrinsic religiosity to be positively associated with flourishing. In contrast, no significant association was found between extrinsic religiosity and flourishing for all models. The widower model found years of education and poverty status to be significantly associated with flourishing, where higher education levels and non-hardcore poverty were significantly related to higher levels of flourishing among widowers. Interestingly, the interaction term, social networks × extrinsic religiosity, was only significant in the overall sample and widow model. This meant that extrinsic religiosity moderated the association between social networks and flourishing among the overall sample (β = −0.126, *p* = 0.002) and widows (β = −0.133, *p* = 0.003), but not among widowers (β = 0.025, *p* = 0.840). Intrinsic religiosity, on the other hand, did not moderate the relationship between social networks and flourishing for all models. Similarly, its interaction term, social networks × intrinsic religiosity, yielded no significant associations in all models. The interaction between social networks and extrinsic religiosity on flourishing among the overall sample is depicted in Figure 1, where the relationship between social networks and flourishing is plotted according to different levels of extrinsic religiosity. Widowed older adults with low levels of extrinsic religiosity reported the lowest levels of flourishing in smaller social networks (equal and above the mean). 

### 3.4. Post-Hoc Analysis Investigating Significance of Difference in Simple Slopes

Extrinsic religiosity was categorized into three groups based on percentiles: low, moderate, and high levels of extrinsic religiosity. The difference in the relationship between social networks and flourishing was then examined based on low (n = 151) and high (n = 163) levels of extrinsic religiosity, respectively [49]. Simple regression analysis showed that social networks significantly predicted flourishing only among widowed older adults with low levels of extrinsic religiosity (β = 0.508, *p* < 0.001). No significant associations were found among those with high levels of extrinsic religiosity (β = 0.016, *p* = 0.841). Next, there was a statistically significant difference (t_312_ = 7.684, *p* < 0.001) in simple slopes between those with low (b = 0.566, SE = 0.079, t = 7.179, *p* < 0.001) and high levels of extrinsic religiosity (b = 0.012, SE = 0.061, t = 0.202, *p* = 0.841). In specific, social networks were strongly correlated with flourishing among widowed older adults with low levels of extrinsic religiosity. 

## 4. Discussion

This study examined the role of intrinsic and extrinsic religiosity as moderators in the relationship between social networks and flourishing among widowed older adults. H_2_A was not supported when findings revealed intrinsic religiosity did not moderate the association between social networks and flourishing. This study also found that social networks and intrinsic religiosity were positively associated with flourishing in all models. Larger social networks were associated with higher levels of flourishing, regardless of intrinsic religiosity levels. The negative relationship between social networks and flourishing in widowed older people concur with a separate study, which found that smaller social networks were associated with lower psychological well-being [54]. The absence of moderating effects is possibly attributable to the importance of both religiosity and social resources in maintaining the proper well-being of widowed older adults. However, widowed older people require social support (such as a large social network) to maintain proper well-being despite strong intrinsic resources (e.g., spirituality). This is further backed by the Biopsychosocial–Spiritual Model, which explains a complex interplay of biological, psychological, social, and spiritual components in mental health [55]. Being widowed is a major life event that involves the loss of a loved one. The experience of losing an intimate partner with whom one shared life experiences, lived with, and confided is often associated with stress [4]. Therefore, having larger social networks can be beneficial during widowhood, as it encourages social interaction, increases social support, and enables the sharing of life experiences, thereby promoting higher levels of subjective well-being [19]. 

H_3_A was supported when extrinsic religiosity was found to moderate the association between social networks and flourishing. To further elaborate, those with low extrinsic religiosity showed steeper associations with social networks and flourishing. These findings reveal that larger social networks are important to maintaining proper well-being among widowed older adults with low extrinsic religiosity. This is likely because individuals with low extrinsic religiosity often lack the social connections required to achieve personal goals [56]. Therefore, having a large social network could serve as a social resource for older adults to cope with bereavement during widowhood. Another possible explanation is poorer social skills and greater perceived support among those with low extrinsic religiosity. Previous studies reported that extrinsic religiosity was positively correlated with social skills but negatively correlated with perceived support [57,58]. As such, poor social skills and high perceived support could have served as factors limiting the expansion of social networks and socialization among widowed older adults, ultimately compromising their well-being.

H_4_A was partially supported when the moderating effects of extrinsic religiosity were significant only among widows and not widowers. However, there were no significant gender variations in the moderating effects of intrinsic religiosity for both models. Gender differences found for the moderating effects of extrinsic religiosity could be attributed to later-life social resources among widows. Malaysian men and women in general had different levels of socialization in adulthood [59]. It was found that older men had higher levels of education than older women. This could be attributed to traditional gender roles, in which men are expected to be the breadwinners of the family and engage in income-generating activities. Meanwhile, women primarily engage in non-income-generating activities such as homemaking and/or childbearing. Consequently, older women had fewer social and financial resources upon the demise of their spouse. Therefore, larger social networks could be an economical and effective social resource to ensure well-being among widows. Nonetheless, the benefits of investing in social relationships during early life can be reaped in later life to buffer losses. However, this study firmly believes that gender differences will diminish over time as women in Malaysia are now equipped with a range of resources, which include equal opportunities in receiving higher education, career advancement, and social networking. 

Several limitations of this study need to be acknowledged. Firstly, the cross-sectional nature of this study made it impossible to establish a causal relationship. Next, the original study was only conducted in Peninsular Malaysia. Therefore, results could not be generalized to widowed older adults residing in East Malaysia. Future studies should perhaps employ the cohort study design and involve all states in Malaysia. In terms of measurement, religiosity was investigated only from a religious orientation perspective. Malaysia is a multi-ethnic, multi-religion country, so the inclusion of religious affiliation may be of importance. Moreover, this study only measured the quantitative aspect of social networks. Qualitative aspects of social networks, such as relationship quality, were not taken into consideration. Therefore, future studies should consider including religious affiliation and relationship quality to measure religiosity and social networks. Furthermore, this study only involved the subgroup of widowed older adults without considering the non-widowed group. Future research should also consider comparing widowed and non-widowed groups to identify if the relationships are unique to the widowed group. 

Nevertheless, this study also has its strengths. Firstly, the sample size used in this study was representative of widowed older adults in Peninsular Malaysia. The distribution of gender across the widowed community-dwelling older adults was of reasonable proportion. Additionally, this study examined flourishing among older adults, a highly under-investigated concept. Gerontologists tend to go with life satisfaction, quality of life, depressive symptoms, and stress to measure positive psychology and well-being. 

## 5. Conclusions

This study revealed that social networks and intrinsic religiosity were associated with higher levels of flourishing in widowed older adults. In terms of healthcare implications, our findings highlight the need for healthcare practitioners or social workers to include social networks and intrinsic religiosity when assessing well-being among widowed older people. Moreover, this study found that extrinsic religiosity moderated the association between social networks and flourishing. The relationship between social networks and flourishing was steeper for those with lower extrinsic religiosity. Lastly, gender variation was present only when extrinsic religiosity moderated the relationship in widows. The findings of this study have implications for community care. Since extrinsic religiosity is associated with making friends while attending worship, this study suggests that older men who cannot participate in religious activities should be encouraged to extend their social networks using other resources. For example, interacting with others through social media and neighborhoods ensures continuous social support and good well-being despite widowhood. This study provides theoretical implications that contribute to the literature on Pargament’s Theory of Religious Coping. The findings suggest that intrinsic religiosity can be a resource for promoting flourishing among widowed older people. Furthermore, this study highlights that the relationship between social networks and flourishing depends on the level of extrinsic religiosity.

## Figures and Tables

**Figure 1 healthcare-11-01300-f001:**
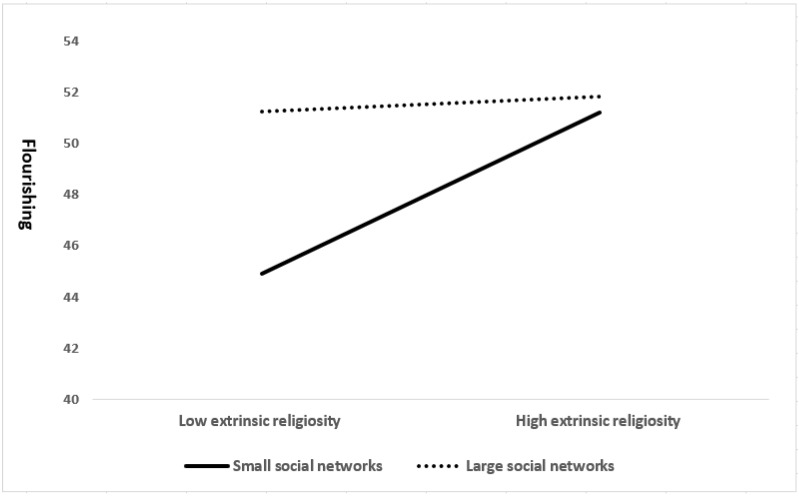
The interaction between social networks and extrinsic religiosity in flourishing.

**Table 1 healthcare-11-01300-t001:** Descriptive analysis of selected independent and dependent variables among older men and women.

		Total (N = 655)	Widower (*n* = 97)	Widow (*n* = 558)	Chi-Square Statistic or *t*-Value	*p*-Value
		*n* (%) or Mean ± SD	*n* (%) or Mean ± SD	*n* (%) or Mean ± SD		
**Age**	**Range: 60–91**	**71.4 ± 6.79**	74.0 ± 7.02	71.0 ± 6.66	4.030 ^a^	**<0.001** ^c^
Year(s) of education	Range: 0–16	3.5 ± 3.53	5.0 ± 3.39	3.3 ± 3.50	4.532 ^a^	**<0.001** ^c^
Employment status	Currently working	93 (14.5)	21 (22.1)	72 (13.2)	5.189 ^b^	**0.023** ^d^
	Currently not working	548 (85.5)	74 (77.9)	474 (86.8)		
Poverty status	Non-hardcore poor	338 (53.7)	59 (62.8)	279 (52.1)	3.625 ^b^	0.057 ^d^
	Hardcore poor	291 (46.3)	35 (37.2)	256 (47.8)		
Multimorbidity status	No multimorbidity	328 (50.1)	50 (51.5)	278 (49.8)	0.098 ^b^	0.754 ^d^
	Multimorbidity	327 (49.9)	47 (48.5)	280 (50.2)		
Living arrangement	Living with others	479 (73.1)	68 (70.1)	411 (73.7)	0.531 ^b^	0.466 ^d^
	Living alone	176 (26.9)	29 (29.9)	147 (26.3)		
Ethnicity	Malay	448 (68.6)	70 (72.2)	378 (68.0)	0.670 ^b^	0.413 ^d^
	Non-Malay	205 (31.4)	27 (27.8)	178 (32.0)		
Flourishing	Range: 8–56	49.8 ± 6.81	50.7 ± 6.41	49.6 ± 6.88	1.451 ^a^	0.147 ^c^
Social networks	Range: 0–30	13.0 ± 6.46	14.2 ± 6.89	12.8 ± 6.36	1.969 ^a^	**0.049** ^c^
Social networks category	No risk of social isolation (social ≥ 13 networks score)	333 (52.0)	51 (53.7)	282 (51.6)	0.134 ^b^	0.714 ^d^
	At the risk of social isolation (social networks score < 13)	308 (48.0)	44 (46.3)	264 (48.4)		
Intrinsic religiosity	Range: 6–30	26.7 ± 3.79	26.8 ± 4.31	26.7 ± 3.70	0.257 ^a^	0.797 ^c^
Extrinsic religiosity	Range: 14–39	26.6 ± 4.67	27.0 ± 4.79	26.5 ± 4.65	0.231 ^a^	0.283 ^c^

Note: *n* = frequency; SD = standard deviation; ^a^ = *t*-value; ^b^ = chi-square statistic; ^c^ = independent sample *t*-test; ^d^ = chi-square test. Significance of bold values = *p* < 0.05.

**Table 2 healthcare-11-01300-t002:** Pearson correlation matrix for study variables.

	1	2	3	4	5	6	7	8	9	10	11	12
1. Sex	-											
2. Age	−0.156 **	-										
3. Education	−0.171 **	−0.368 **	-									
4. Employment status	0.090 *	0.114 **	−0.063	-								
5. Ethnicity	−0.032	−0.045	−0.017	−0.035	-							
6. Living arrangement	−0.028	0.017	−0.007	−0.032	−0.043	-						
7. Poverty status	0.076	0.129 **	−0.275 **	0.007	0.066	0.092 *	-					
8. Multimorbidity status	0.012	−0.022	0.022	0.154 **	−0.125 **	0.022	0.014	-				
9. Social networks	−0.078 *	−0.019	0.072	−0.057	0.020	0.027	−0.092 *	−0.027	-			
10. Intrinsic religiosity	−0.010	−0.067	0.015	−0.046	0.402 **	0.004	0.157 **	0.009	0.155 **	-		
11. Extrinsic religiosity	−0.043	−0.008	0.058	−0.093 *	0.284 **	0.103 *	0.159 **	−0.052	−0.087 *	0.370 **	-	
12. Flourishing	−0.058	−0.097 *	0.091 *	−0.045	0.201 **	0.018	0.053	−0.072	0.238 **	0.418 **	0.222 **	-

Note: * *p*-value < 0.05, ** *p*-value < 0.01.

**Table 3 healthcare-11-01300-t003:** The moderating roles of intrinsic and extrinsic religiosity in the relationship between social networks and flourishing among older men and women.

Variables	(1) Flourishing—Overall			(2) Flourishing—Widower			(3) Flourishing—Widow		
	*β*	SE	*p*-Value	*Β*	SE	*p*-Value	*β*	SE	*p*-Value
Sex (1—widow, 0—widower)	−0.043	0.735	0.254	N/A	N/A	N/A	N/A	N/A	N/A
Age	−0.061	0.042	0.141	−0.045	0.094	0.673	−0.055	0.047	0.219
Year(s) of education	0.069	0.082	0.096	0.248	0.229	**0.039**	0.057	0.089	0.204
Employment status (1—currently not working, 0—currently working)	0.008	0.724	0.820	−0.014	1.405	0.883	0.008	0.836	0.842
Ethnicity (1—Malay, 0—non-Malay)	0.018	0.597	0.658	0.030	1.438	0.770	0.018	0.663	0.689
Living arrangement (1—living alone, 0—living with others)	−0.006	0.561	0.875	−0.023	1.233	0.806	−0.007	0.628	0.859
Poverty status (1—hardcore poor, 0—non-hardcore poor)	0.030	0.531	0.440	0.251	1.548	**0.045**	0.013	0.576	0.760
Multimorbidity status (1—multimorbidity, 0—non-multimorbidity)	−0.065	0.503	0.079	−0.180	1.139	0.058	−0.048	0.561	0.232
Social networks	0.206	0.040	**<0.001**	0.203	0.090	**0.045**	0.206	0.045	**<0.001**
Intrinsic religiosity	0.323	0.080	**<0.001**	0.356	0.168	**0.004**	0.317	0.091	**<0.001**
Extrinsic religiosity	0.075	0.060	0.067	0.114	0.135	0.291	0.070	0.067	0.121
Social networks × intrinsic religiosity	0.044	0.011	0.299	−0.115	0.025	0.350	0.059	0.013	0.208
Social networks × extrinsic religiosity	−0.126	0.010	**0.002**	0.025	0.032	0.840	−0.133	0.011	**0.003**

Note: 0 = reference group; 1 = non-reference group; *β* = standardized beta coefficient; SE = standard error. Significance of bold values = *p* < 0.05.

## Data Availability

The datasets used and/or analyzed during the current study are available from the corresponding author upon reasonable request.

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
