# Peer review of "The Moderating Roles of Intrinsic and Extrinsic Religiosity on the Relationship between Social Networks and Flourishing: A Study on Community-Dwelling Widowed Older Adults in Malaysia"

_healthcare, 2023, doi:10.3390/healthcare11091300_

Round 1

Reviewer 1 Report

Dear researcher(s), you are addressing an important and meaningful gap. Your paper is well-written and it has some important results, and if you edit your paper it can be much more effective. Here some humble suggestions to improve the paper, I would do the following to strengthen the paper. I have enjoyed reading the paper and am looking forward to seeing the paper published. You could increase the effect of your paper with some more recent studies suggested below or any other studies and not using the suggested ones.

Reviewer 2 Report

Thank you very much for the opportunity to review this manuscript. I believe that it deals with a relevant topic and a population that merits this type of research. However, I detect major limitations that I suggest be reviewed in depth:

Introduction

I think that the introduction should be restructured and its development should be clearer. The arguments in the introduction should be closely linked to the central objective or thesis of this paper: The relationship between social networks and well-being in widowed older adults. I believe that this is not achieved in the current version. My suggestion is that the authors focus on the central thesis of the relationship between social relations and well-being in widowed older adults, and from there argue for the other hypothesised relationships that are considered relevant (religiosity and gender).

At the end of the introduction, some hypotheses are put forward. Here I note the following limitations:

1-     The hypotheses are imprecise. I suggest specifying the hypothesised direction and intensity of the associations.

2-     It is necessary to incorporate a hypothesis that summarises the main relationships (including directionality and intensity) that will be tested later: Social Networks and Well-being.

Methods:

I see two major limitations to the method:

1-     For the measurements of social networks, religiosity, and flourishing, evidence of construct validity (dimensionality-internal structure) in the target population (widowed older adults) must be shown. In addition, given that the aim is to make comparisons by gender, evidence of invariance of the measurement models (internal structure) as a function of gender must also be shown. All this should be worked out in depth, detailing the what, how and why of each psychometric decision taken for these purposes. This is why I suggest that in addition to pointing out the main results of these psychometric analyses in the body of the manuscript, an appendix with details of the procedure followed should be included (see https://psyarxiv.com/pkc7q).

2-     Further regression analyses should be consistent with the results of these previous psychometric analyses. For example, if multidimensional structures are evident in the measurements, this multidimensionality should be incorporated accordingly in the subsequent regression analyses. In addition, I suggest that the authors work from a structural equation modelling (SEM) approach, since it makes it possible to incorporate latent variables and control for measurement error. They can perform these analyses in R (using the Lavaan package), a free and open access software that also makes it possible to make the analysis procedures carried out more transparent. Regarding the latter, I also suggest incorporating in supplementary material the syntaxes or steps followed to make transparent the specific data analysis procedure followed by the authors.

I hope my comments will be useful to the authors and help them to achieve better reporting. My comments are with the utmost respect for the effort and work done by the authors, and I have tried to express myself with great respect for them. I wish them all the best in their work. 

Reviewer 3 Report

The idea of the research is minimalist, which is the role of internal and external religiosity, but despite that, the researchers tried to study it systematically, but now

1- The idea that the research was applied (2012-2013) and its results I expect will differ now in 2023

2- What is meant by wave I

3- where describe the psychometric properties of the scales

Round 2

Reviewer 2 Report

Thank you for this opportunity to review the manuscript again. Unfortunately, the authors have not been able to address one of the major limitations I pointed out in the first round of review. Specifically, in order to establish the validity and comparability of the measurements of social networks, religiosity, and flourishing among the target population (widowed older adults), evidence of construct validity (dimensionality-internal structure) and measurement invariance by gender must be demonstrated. The authors' response to this issue is inadequate, as the importance of evaluating measurement model fit goes beyond the scope of psychometric studies, and is critical for any study that relies on these models for subsequent data analysis and interpretation.
